# SRC-1 controls growth cone polarity and protrusion with the UNC-6/Netrin receptor UNC-5 in *Caenorhabditis elegans*

**Snehal S. Mahadik, Emily K. Burt◉, Erik A. Lundquist◉***

Program in Molecular, Cellular and Developmental Biology, Department of Molecular Biosciences, University of Kansas, Lawrence, KS, United States of America

\* erikl@ku.edu

## Abstract

The Polarity/Protrusion model of UNC-6/Netrin function in axon repulsion does not rely on a gradient of UNC-6/Netrin. Instead, the UNC-5 receptor polarizes the VD growth cone such that filopodial protrusions are biased to the dorsal leading edge. UNC-5 then inhibits growth cone protrusion ventrally based upon this polarity, resulting in dorsally-biased protrusion and dorsal migration away from UNC-6/Netrin. While previous studies have shown that UNC-5 inhibits growth cone protrusion by destabilizing actin, preventing microtubule + end entry, and preventing vesicle fusion, the signaling pathways involved are unclear. The SRC-1 tyrosine kinase has been previously shown to physically interact with and phosphorylate UNC-5, and to act with UNC-5 in axon guidance and cell migration. Here, the role of SRC-1 in VD growth cone polarity and protrusion is investigated. A precise deletion of *src-1* was generated, and mutants displayed unpolarized growth cones with increased size, similar to *unc-5* mutants. Transgenic expression of *src-1(+)* in VD/DD neurons resulted in smaller growth cones, and rescued growth cone polarity defects of *src-1* mutants, indicating cell-autonomous function. Transgenic expression of a putative kinase-dead *src-1(D831A)* mutant caused a phenotype similar to *src-1* loss-of-function, suggesting that this is a dominant negative mutation. The D381A mutation was introduced into the endogenous *src-1* gene by genome editing, which also had a dominant-negative effect. Genetic interactions of *src-1* and *unc-5* suggest they act in the same pathway on growth cone polarity and protrusion, but might have overlapping, parallel functions in other aspects of axon guidance. *src-1* function was not required for the effects of activated *myr::unc-5*, suggesting that SRC-1 might be involved in UNC-5 dimerization and activation by UNC-6, of which *myr::unc-5* is independent. In sum, these results show that SRC-1 acts with UNC-5 in growth cone polarity and inhibition of protrusion.

## Introduction

Growth cones at the distal tip of a growing axon sense and respond to extracellular cues in the organism that provide axon guidance information. Growth cones are dynamic structures that

**Data Availability Statement:** Whole-genome sequencing data for src-1(cj293) can be found in the Sequence read Archive, BioProject ID PRJNA973980. C. elegans strains used in this work

can be found at the Caenorhabditis Genetics Center (https://cgc.umn.edu/). The sequences of plasmids can be found in Supplemental Material.

**Funding:** PI EAL, R03NS114554, National Institute of neurological Disorders and Stroke PI EAL, R01NS115467, National Institute of neurological Disorders and Stroke PI not an author, P30GM145499, National Institute of General Medical Sciences PI not an author, P20GM103418, National Institute of General Medical Sciences PI not an author, P40OD010440, National Institutes of Health EAL Internal University of Kansas research funds There was no additional external funding received for this study.

**Competing interests:** The authors have declared that no competing interests exist.

are made up of filopodial and lamellopodial protrusions and both are important for proper axon outgrowth and guidance [1–3]. The interaction between the receptors on the surface of the growth cones, and guidance cues in the surrounding environment affects axon guidance by altering growth cone protrusion.

In *C. elegans*, UNC-6/Netrin is a bi-functional signaling molecule that acts as both an attractant and repellant in axon guidance of VD/DD neurons and mediates dorsal ventral circumferential migration of growth cones axons with the help of receptors UNC-5 and UNC-40 [4–8]. Previous work using time-lapse imaging of growth cones in *C. elegans* showed that the attractive UNC-6/Netrin receptor UNC-40/DCC homodimer stimulates protrusion of growth cones, and the repulsive netrin receptor UNC-5/UNC-40 heterodimer inhibits the protrusion of growth cones [4]. UNC-6/Netrin and the UNC-5 receptor polarize the growth cone such that protrusive activity is asymmetric across the growth cone, with dorsal stimulation of protrusion and ventral inhibition of protrusion. This results in directed growth cone migration away from UNC-6/Netrin (the Polarity/Protrusion model) [9].

Two orthologs of Src family kinases identified in *C. elegans* are SRC-1 and SRC-2/kin22 [10]. SRC-1 is a non-receptor tyrosine kinase, structurally and functionally similar to Src in humans, and controls early embryonic polarity and endoderm specification [11]. SRC-1 also regulates gonadal distal tip cell migration and axon guidance [12]. SRC-1 physically interacts with the cytoplasmic domain of UNC-5 via its SH2 domain and possibly SH3 domain, and acts with UNC-5 in axon guidance [13]. Indeed, UNC-5 is phosphorylated on tyrosines in an UNC-6/Netrin and SRC-1 dependent manner [13, 14], and mutation of tyrosine at 482 of UNC-5, one of the phosphorylated residues, severely affects axon guidance, DTC migration, and growth cone protrusion [15, 16]. In the context of distal tip cell migration, SRC-1 mediates UNC-5 signaling by acting downstream of UNC-5 [12]. Together, these data strongly indicate that SRC-1 acts with UNC-5 in UNC-6/Netrin-mediated axon guidance, and that SRC-1 phosphorylation of UNC-5 is important in this role.

To further explore the UNC-5/SRC-1 interaction in the context of the Polarity/Protrusion model, VD growth cones of *src-1* mutants were analyzed. If UNC-5 and SRC-1 acted in the same signaling pathway, it was expected that *src-1* mutant growth cones would resemble those of *unc-5* mutants (large, overly-protrusive growth cones with loss of dorsal polarity of filopodial protrusion). A precise deletion of the entire *src-1* gene was produced using genome editing. *src-1(lq185)* resulted in sterile adults which displayed unpolarized VD growth cones with increased area and filopodial length, similar to *unc-5* loss-of-function. Thus, loss-of-function of *src-1* resembled loss-of-function of *unc-5*, suggesting that they act in the same process in growth cone morphology and axon guidance.

In the existing *src-1(cj293)* deletion mutant [11], growth cones were unpolarized, similar to *unc-5*. Surprisingly, *src-1(cj293)* growth cones were smaller and less protrusive than wild type, the opposite of *unc-5* and *src-1(lq185)*. *src-1(cj293)* was reported to remove the SH2 domain, part of the kinase domain, and to result in a frame shift [11]. The sequence reported in here shows that *cj293* results in deletion of the SH2 domain and a regulatory region of the kinase domain, but does not introduce a frame shift. Thus, *src-1(cj293)* could encode a SRC-1 molecule lacking the SH2 domain and a regulatory region of the kinase domain. This also suggested that *src-1(cj293)* might be an activated allele of *src-1*. Alternatively, *src-1(cj293)* does not affect the coding region of the *src-1D* isoform, which is composed of the last two exons encoding the kinase domain. Expression of either of these molecules in the absence of full-length isoforms could result in the gain-of-function effect of *src-1(cj293)*.

Transgenic expression of *src-1* resulted in VD growth cones with reduced protrusion, and rescued *src-1(lq185)*, indicating that *src-1* acts cell-autonomously in the growth cone. Mutation of the catalytic aspartate 381 (D381A) abolished rescue and in fact led to VD growth cones

with excess protrusion similar to *src-1* loss-of-function. This suggests that the kinase-dead D381A mutation is a dominant negative. Indeed, a D381A genome edit, *src-1(syb7248)*, resulted in embryonic lethality with dominant increase in VD growth cone protrusion and loss of polarity, consistent with the D381A mutation being a dominant negative.

Finally, genetic interactions between *unc-5* and *src-1* were consistent with interaction in the same genetic pathway, although some redundant roles axon guidance were also discovered. This suggests that SRC-1 might also act in parallel to UNC-5 in axon guidance. In sum, results presented here, along with previously-published studies, indicate that SRC-1 acts with UNC-5 in axon guidance and VD growth cone polarity and protrusion.

## Results

### A precise deletion allele of the *src-1* locus

The extant *src-1(cj293)* deletion results in maternal-effect lethality with defects in early embryonic polarity and endoderm specification [11]. The *cj293* mutation was reported to be a 4.5kb deletion that removed coding region for the SH2 domain and part of the kinase domain, and to result in a frame shift [11].

To determine the exact the *cj293* sequence, *src-1(cj293)* mutants were subject to whole-genome sequencing. A 5381-bp deletion in *src-1* was found (Fig 1A and S1 File), which had breakpoints in introns two and four, and removed coding exons three and four (the same coding region reported removed in [11]). However, if the fusion intron consisting of the 5' splice site of exon two and 3' splice site of exon four was removed from transcripts, and exons two and five were spliced together, the *src-1* open reading frame would be retained. The resulting *src-1(cj293)* gene product would lack the SH2 domain and the first 45 codons of the SH1 (kinase) domain, including lysine 290 in the regulatory C-loop of the kinase domain (Fig 1B) [11, 13, 17]. The predicted molecule would have an intact SH3 domain and a majority of the SH1 (kinase) domain, including the catalytic aspartate 381 (Fig 1B). The *cj293* in-frame deletion reported here has the potential to encode a SRC-1 molecule lacking the SH2 domain and a regulatory region of the kinase domain, but with intact SH3 and catalytic kinase domains, and might not represent a complete loss of *src-1* function. Using RT-PCR, no cDNA representing the predicted splice in *cj293* was identified, suggesting that this splice might not occur. The *cj293* deletion does not affect the coding region of the predicted *src-1D* isoform, which is composed of the last two exons encoding the kinase domain (Fig 1A).

To generate a *src-1* null mutation, CRISPR/Cas9 genome editing was used to create a precise deletion of the *src-1* gene lacking nearly the entire coding potential except for the first seven and last two codons, called *src-1(lq185)* (Fig 1A). *src-1(lq185)* homozygotes grew to sterile adults that produced no apparent embryos. The effects of *src-1(lq185)* on embryonic polarity and endoderm specification have not been determined.

### *src-1* controls VD/DD motor axon guidance and VD growth cone morphology

*src-1* was previously shown to affect gonadal cell migration, neuronal migration, and axon pathfinding [12]. Furthermore, *src-1* interacts with the UNC-6/Netrin receptor UNC-5 in axon guidance [13]. The VD and DD neurons are both labeled with the *Punc-25::gfp* transgene despite the fact that the DD axons extend embryonically and the VD axons extend in the early L2 larval stage. *src-1(lq185)* and *src-1(cj293)* mutants both displayed similar levels of defects in VD/DD axon guidance, including axon wandering, ectopic branching, and failure to reach the dorsal nerve cord (Figs 2A and 3A–3C).

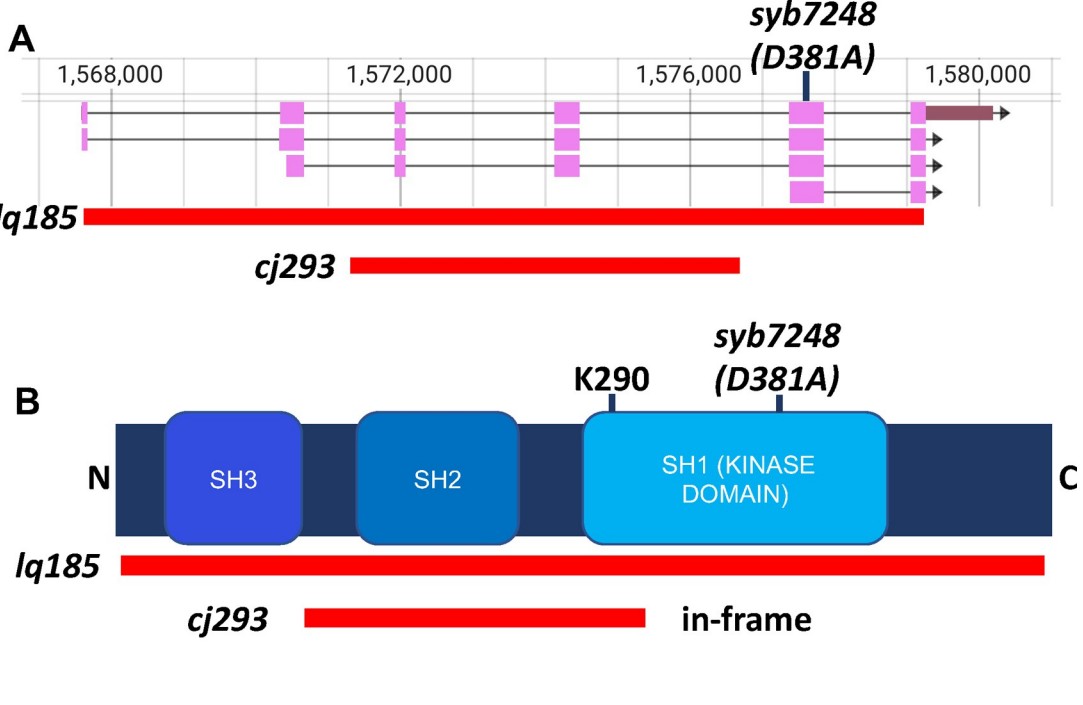

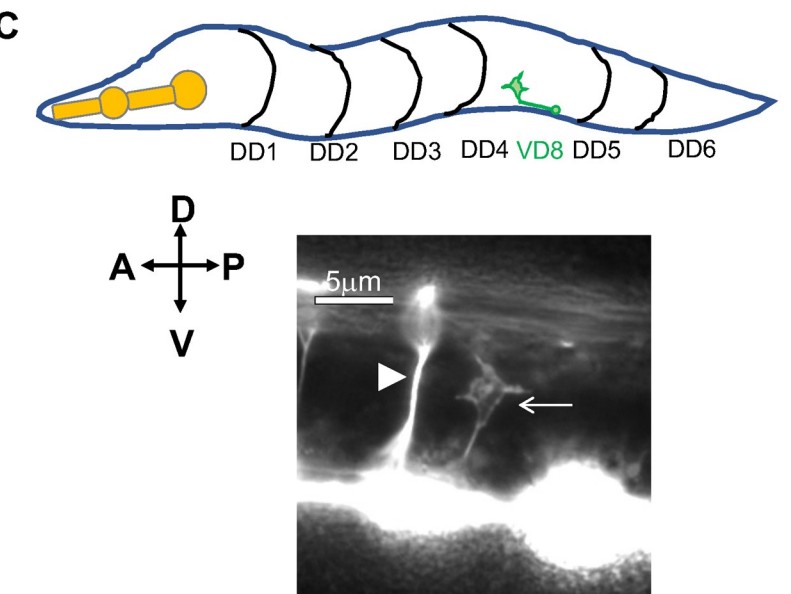

**Fig 1. *src-1* mutations and VD growth cone outgrowth.** A) *src-1* gene structure (from Wormbase) and mutations are shown. Red bars indicate the extent of deletions. B) Domains of SRC-1 family are indicated (Src-homology SH3 and SH2 domains, and SH1 kinase domain with catalytic residue at 381 aspartic acid (D381)). Red bars indicate the coding region that is deleted in respective mutations, and the *syb4278 D381A* catalytic residue mutation is indicated. C) Schematic of an early L2 larva of showing the DD motor neuron commissures (black lines) and a VD8 growth cone undergoing dorsal outgrowth (green). A fluorescent micrograph of an early L2 larva with *unc-25::gfp* showing a wild-type VD growth cone undergoing dorsal outgrowth (arrow). The arrowhead points to the DD4 commissural axon. The scale bar represents 5μm. Anterior is to the left, and dorsal is up.

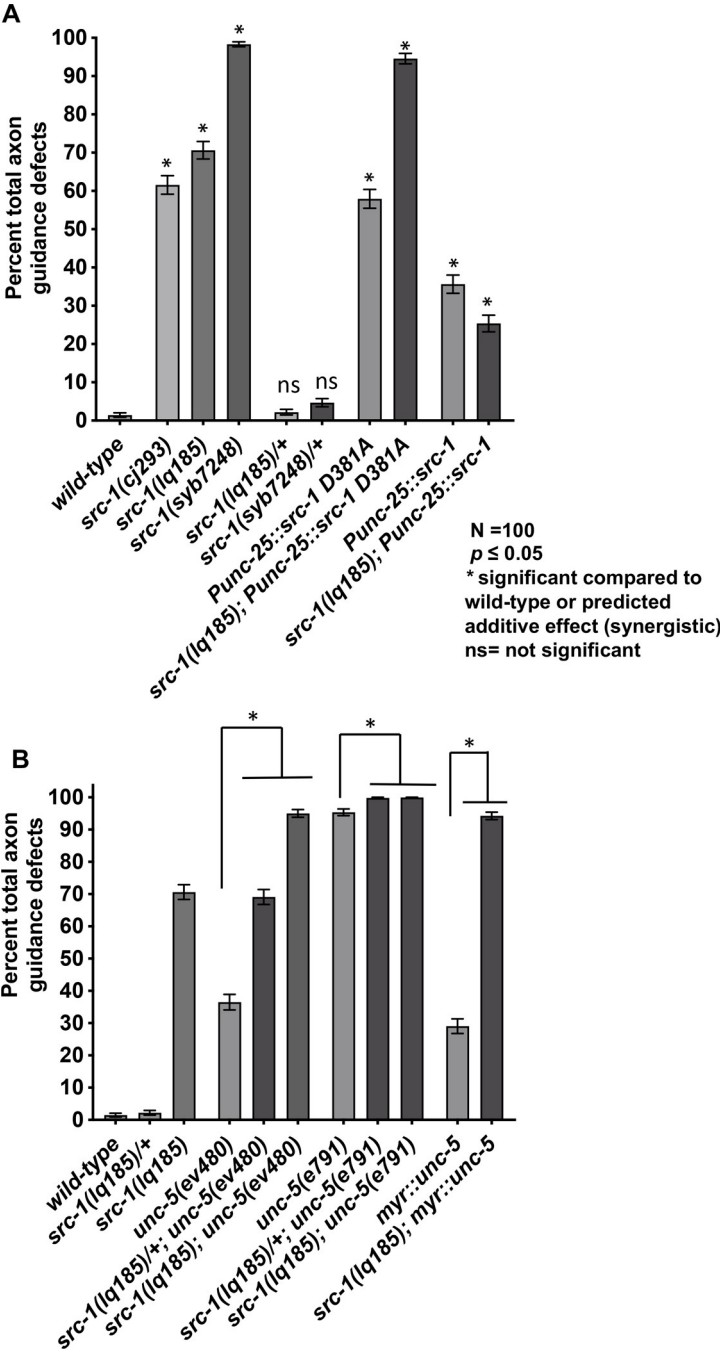

**Fig 2. Quantification of VD/DD axon guidance defects.** A) Total VD/DD axon guidance defects (Y axis) in different genotypes (x axis) are shown, including axon wandering, branching, and failure to extend past the lateral midline (see Materials and Methods). Error bars represents 2x standard error of the proportion. Significance of difference compared to wild-type (asterisks) was determined using Fisher's exact test ($p < 0.05$). ns = not significant. 100 animals (1600) commissures) were scored (see Materials and Methods). B) The graph is displayed as described in (A). The predicted additive effect of double mutants calculated by the formula p1 + p2 −(p1p2), where p1 and p2 are the proportions of axon defects in each single mutant.

VD growth cone morphology was analyzed in *src-1* mutants as previously described [4, 9, 18–21]. In early L2 larvae, the VD growth cones migrate commissurally from the ventral to dorsal nerve cord. During commissural extension between the lateral midline and the ventral and

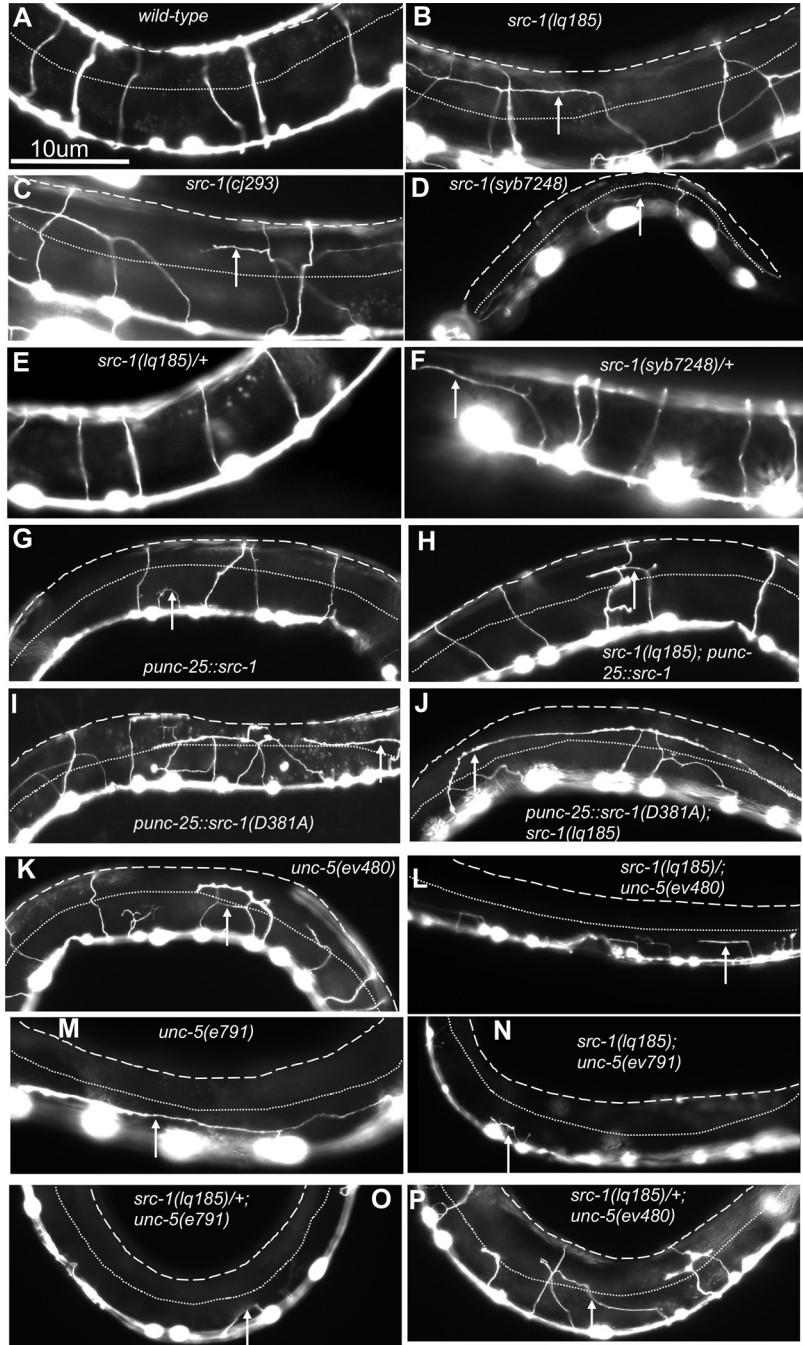

**Fig 3. VD/DD axon guidance defects.** Fluorescent micrographs of L4 animals of indicated genotypes with the *Punc-25::gfp* transgene *juIs76* expressed in the VD/DD neurons. *Punc-25::src-1* and *Punc-25::src-1(D381A)* represent transgenic lines of *src-1* expression in the VD/DD neurons. Dorsal is up and anterior left. The approximate lateral midline is indicated with a dotted white line, and the dorsal nerve cord by a dashed white line. White arrows indicate axon guidance defects in each genotype. Scale bar in (A) represents 10μm.

dorsal nerve cords, wild-type VD growth cones display a robust growth cone body and filopodial protrusions (Fig 1C). Wild-type VD growth cones display an average growth cone area of 4.6μm$^2$, and an average filopodial length of 0.95μm (Fig 4A, 4B and 4D). Furthermore, growth cone filopodial protrusions were dorsally biased in the direction of growth (Fig 4C and 4D).

The *src-1(lq185)* precise deletion mutants displayed significantly increased growth cone area compared to wild-type, but did not significantly affect filopodial length (Fig 4A, 4D and 4E), similar to *unc-5* mutants. In contrast, *src-1(cj293)* mutants displayed significantly reduced growth cone area and filopodial length compared to wild-type, the opposite of *unc-5* and *src-1 (lq185)*. The opposing effects of *cj293* and *lq185* on growth cone area suggests that *cj293* is not a null, complete loss-of-function mutant and that it might result in SRC-1 overactivity. Both *src-1(cj293)* and *src-1(lq185)* displayed unpolarized growth cones with loss of dorsal bias of filopodial protrusion (Fig 4C, 4E and 4F). In sum, analysis of *src-1(lq185)* complete loss-of-function suggests that SRC-1 is required to limit VD growth cone area and protrusion, and is also required for growth cone dorsal polarity of protrusion. Possibly, the SH2 domain, missing in *src-1(cj293)*, has an inhibitory effect on SRC-1 activity, resulting in reduced growth cone protrusion. Alternatively, expression of *src-1D*, unaffected by the *cj293* mutation and encoding only the kinase domain (Fig 1A), might result in a gain-of-function phenotype in the absence of the full-length isoform. A full analysis of *cj293* as a potentially activated allele is beyond the scope of this report, and experiments will focus on *src-1(lq185)* loss-of-function.

## SRC-1 acts cell-autonomously in axon pathfinding and growth cone morphology

Transgenic expression of the wild-type *src-1(+)* coding region was driven in VD/DD motor neurons using the *unc-25* promoter. *Punc25::src-1(+)* expression significantly rescued VD/DD axon guidance defects of *src-1(lq185)* (Figs 2, 3G and 3H). It also rescued the increased growth cone area and dorsal polarity of filopodial protrusion of *src-1(lq185)* (Fig 5).

In a wild-type background, *Punc-25::src-1(+)* transgenic expression resulted in some axon guidance defects, albeit weaker than *src-1(lq185)* (Figs 2 and 3G). Furthermore, it resulted in growth cones with smaller area and shortened filopodial length compared to wild-type (Fig 5A–5C and 5E). Transgenic expression of *src-1(+)* might result in too much SRC-1 activity, and thus smaller growth cones with reduced filopodial protrusions, consistent with a role of SRC-1 in inhibiting growth cone protrusion. Dorsal polarity of filopodial protrusion was unaffected by *src-1(+)* transgenic expression (Fig 5C). Overall, these results indicate that *src-1* acts cell autonomously in VD/DD axon guidance and regulation of VD growth cone polarity and protrusion. However, transgenic expression of *src-1(+)* caused significant defects on its own that are consistent with *src-1* overactivity.

## *src-1* and *unc-5* might act together in the same pathway

SRC-1 is known to physically interact with and phosphorylate UNC-5, suggesting that UNC-5 and SRC-1 act together in a common genetic pathway. *unc-5(e791); src-1(lq185)* null double mutants showed a slight but significant increase in VD/DD axon guidance defects compared *unc-5(e791)* (95% to 100%), with a near complete failure of any axons extending commissurally (Figs 2B and 3N). VD/DD axon guidance defects in hypomorphic *unc-5(ev480)* were significantly weaker than those in the null allele *unc-5(e791)* (Figs 2 and 3K–3M). *unc-5(ev480); src-1(lq185)* double mutants displayed axon guidance defects similar to the *unc-5(e791)* null mutant (Fig 2B). Furthermore, *src-1(lq185)* dominantly enhanced *unc-5(ev480)* to the level of *unc-5(e791)* null, and slightly enhanced *unc-5(e791)* axon guidance defects (*src-1(lq185)/+* heterozygous) (Figs 2B, 3O and 3P). Dominant enhancement is consistent with activity in the same genetic pathway. These results are consistent with UNC-5 and SRC-1 acting in the same pathway in axon guidance. However, the slight enhancement of the *unc-5(e791)* null suggests they also might act in parallel pathways.

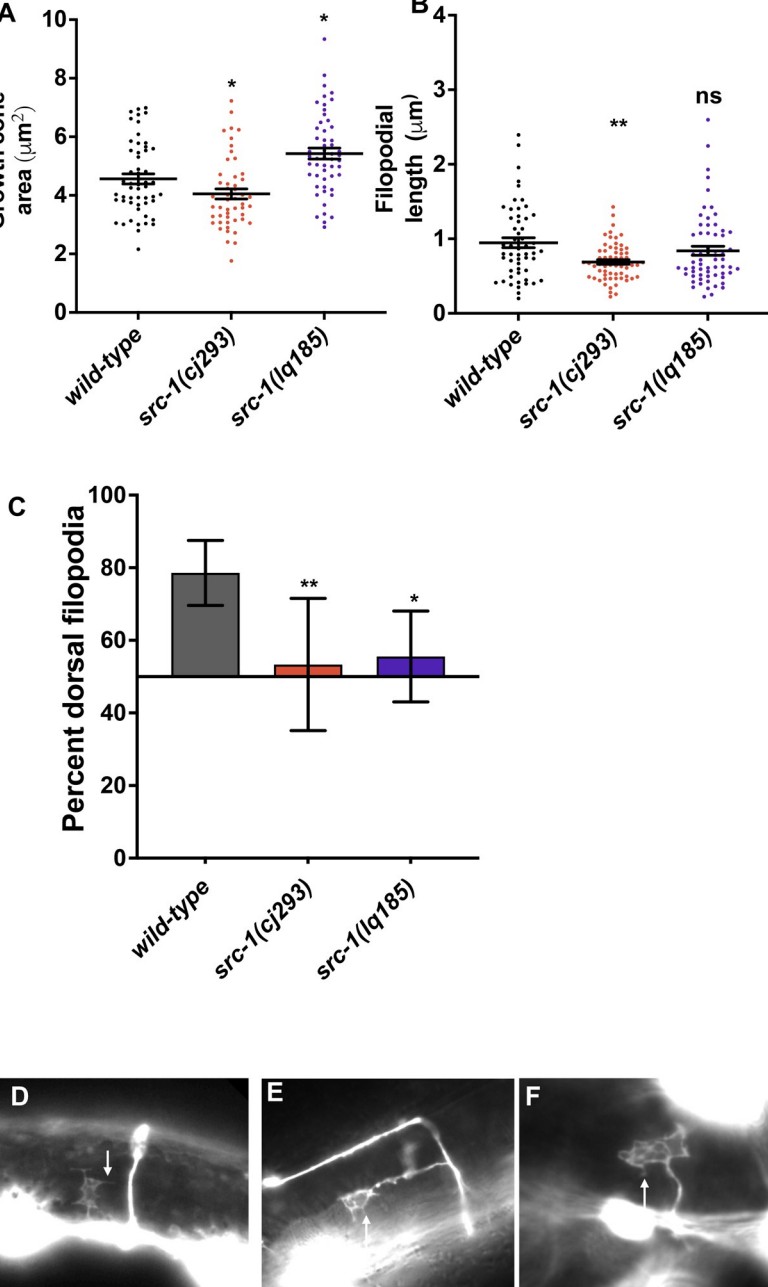

**Fig 4. VD growth cone morphological analysis in *src-1* mutants.** A) The graph shows the growth cone area in μm$^2$ (Y axis) in different genotypes (X axis). Each point represents a measurement from a single growth cone. Error bars indicate standard error of the mean. Two-sided *t*-tests with unequal variance were used to determine the significance between wild-type and mutants. Single asterisks (*) indicate significance at $p < 0.05$. Double asterisks (**) indicate significance at $p < 0.001$. ns = not significant. At least 50 growth cones were scored in each genotype. B) A graph indicating filopodial length in μm (Y axis) in different genotypes (X axis). Each point represents the measurement of a single filopodium. Statistical analysis is as described in (A). C) A graph indicating the percent of dorsally directed filopodial protrusions in VD growth cones (Y axis) in different genotypes. The Y axis is set at 50%, such that bars extending above the Y axis represents above 50%, and bars that extends below represents below 50%. Significance between wild-type and mutants was determined by Fisher's exact test. Error bar represents 2x standard error of proportion. At least 50 growth cones os each genotype were analyzed. Single asterisks (*) indicates the significant $p < 0.05$ double asterisks (**) indicate significant $p < 0.001$. (D-F) Fluorescence micrographs of wild-type and mutant VD growth cones expressing *Punc-25::gfp*. (D, E, F). Arrows point to filopodial protrusions. Dorsal is up; anterior is left. The scale bar in (D) represents 5μm.

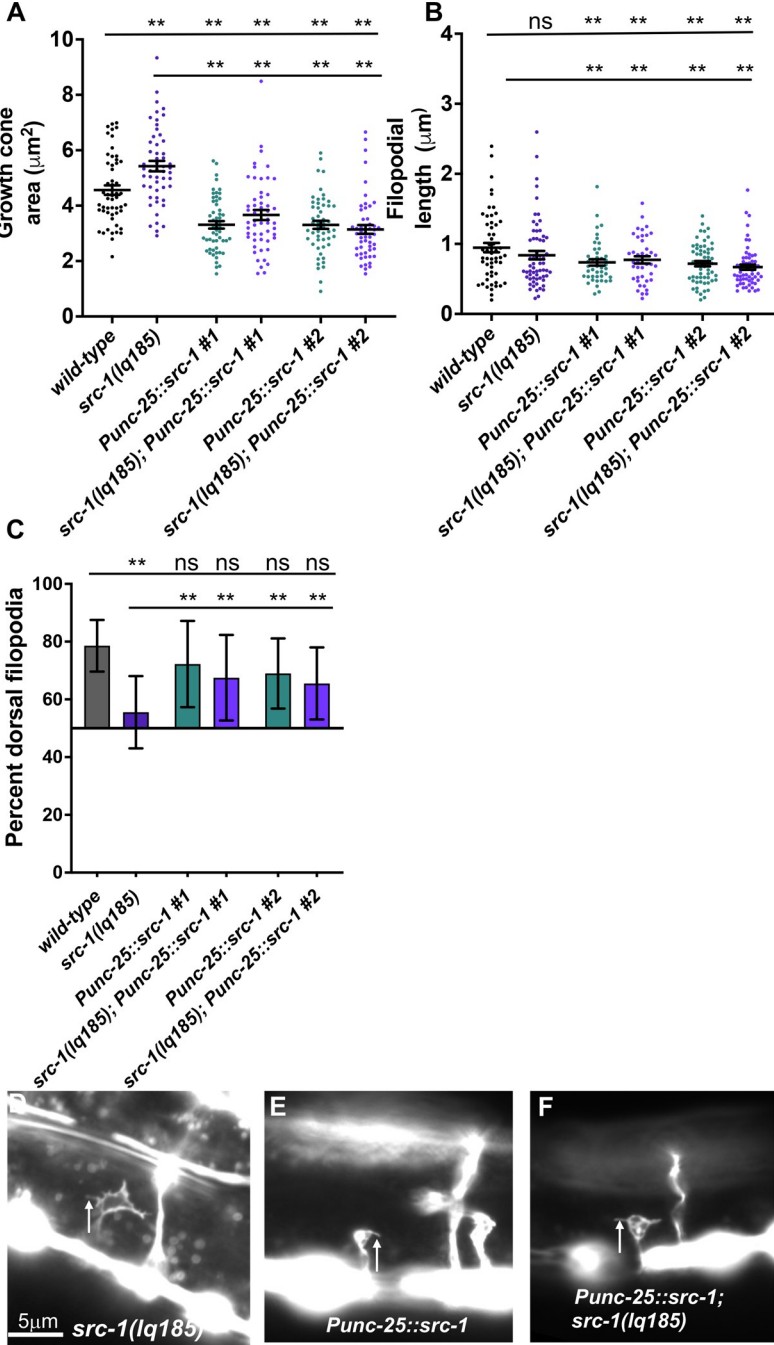

**Fig 5. VD growth cone morphological analysis of *src-1* transgenic expression.** The graphs and statistical analyses in (A-C) are as described in Fig 4. *Punc-25*::*src-1 #1* and *#2* indicate two independent lines of *src-1* expression in the VD/DD neurons. At least 50 growth cones of each genotype were analyzed. The lines indicate the genotypes analyzed for significance, with the left end of the line indicating the genotype to which significance was measured (i.e. wild-type and *src-1(lq185)*). D-F) Fluorescence micrographs of mutant VD growth cones of indicated genotypes expressing *Punc-25*:: *gfp*. Arrows point to filopodial protrusions. Dorsal is up; anterior is left. The scale bar in (D) represents 5μm.

VD growth cones of *unc-5(e791); src-1(lq185)* null double mutants could not be scored because none extended visibly out of the ventral nerve cord. *unc-5(ev480)* is a hypomorphic allele which retained some activity of *unc-5* and displayed a weaker Unc phenotype than the

null alleles [15, 16, 22]. VD growth cones in *unc-5(ev480)* displayed loss of dorsal polarity of protrusion and increased area and filopodia length similar to the *unc-5(e791)* null mutant (Fig 5A–5D) [21]. *unc-5(ev480); src-1(lq185)* VD growth cones resembled those of single mutants (Fig 6). *Src-1(lq185)/+* also did not enhance *unc-5(e791)* (Fig 5). This lack of enhancement is consistent with UNC-5 and SRC-1 acting in the same pathway. *src-1(lq185)/+* significantly decreased growth cone area of *unc-5(ev480)*. This could indicate that SRC-1 has complex interactions with UNC-5 that are not completely understood. For example, src-1 kinase activity could have both positive and negative effects by acting on more than one target.

In sum, the bulk of genetic interactions described here are consistent with SRC-1 and UNC-5 acting together in a common genetic pathway, consistent with the finding that SRC-1 physically interacts with the cytoplasmic region of UNC-5 and phosphorylates it [13]. However, slight enhancement of axon guidance defects suggests that they might also have some minor roles in parallel pathways.

### Activated MYR::UNC-5 does not require SRC-1 for activity

MYR::UNC-5 represents a constitutively-active version of UNC-5, and causes reduced growth cone protrusion [18] (Fig 6A, 6B and 6F). *src-1(lq185); myr::unc-5* double mutants resembled *myr::unc-5* alone, with reduced growth cone area and filopodial length compared to *wild-type* (Fig 6A, 6B and 6I). Indeed, growth cone area was further reduced compared to *myr::unc-5* alone (Fig 6A). There were no significant changes in growth cone polarization of protrusion in double mutants (Fig 6C). These results indicate that SRC-1 is not required for the function of MYR::UNC-5. This could mean that they act in separate pathways. Alternatively, SRC-1 acts upstream of MYR::UNC-5 in the same pathway. In other words, SRC-1 could be required for UNC-5 dimerization with itself or UNC-40, which is not required by MYR::UNC-5.

### *src-1(D381A)*, a predicted kinase-dead mutant, is dominant negative

As shown in Fig 5, transgenic expression of *src-1(+)* in VD/DD neurons rescued *src-1(lq185)* excess growth cone protrusion, consistent with SRC-1 acting cell-autonomously. In a wild-type background, *src-1(+)* transgenic expression inhibited growth cone protrusion, suggesting that SRC-1 overactivity inhibited growth cone protrusion. To determine if the kinase activity of SRC-1 was required for these effects, the catalytic aspartic acid 381 (D381) was mutated to alanine in the *Punc-25::src-1* transgene. In a wild-type background, *src-1(D381A)* expression caused strong axon guidance defects (Figs 2A and 3I). It also caused an increased growth cone area, and loss of dorsal polarity of filopodial protrusion, similar to *src-1(lq185)* mutants (Fig 7). Filopodial length was unaffected. This suggests that *src-1(D381A)* might act as a dominant-negative for *src-1* function. Indeed, *src-1(D831A)* failed to rescue axon guidance defects, growth cone area, and dorsal polarity of filopodial protrusion of *src-1(lq185)* mutants (Fig 7). Axon guidance defects were enhanced (Figs 2A and 3J).

D381 was mutated to alanine in the endogenous *src-1* gene using CRISPR/Cas9 genome editing, called *src-1(syb7248)*. Homozygous *src-1(syb7248)* arrested in the L2 larval stage, earlier than the *src-1(lq185)* knockout mutants, which survived as sterile adults. DD axons were visible in *src-1(syb7248)* mutants and displayed severe axon guidance defects (Figs 2A and 3D). No VD growth cones were apparent in *src-1(syb7248)* homozygotes and so could not be scored. These results suggest that *src-1(syb7248)* is also a dominant-negative mutant.

To test if *src-1(syb7248)* is dominant, heterozygotes were analyzed. *src-1(syb7248)/+* displayed no significant VD/DD axon guidance defects (Figs 2A and 3F). However, this did not differ significantly from wild-type. However, *src-1(syb7248)/+* displayed VD growth cones with significantly increased area and significantly longer filopodia compared to wild-type (Fig

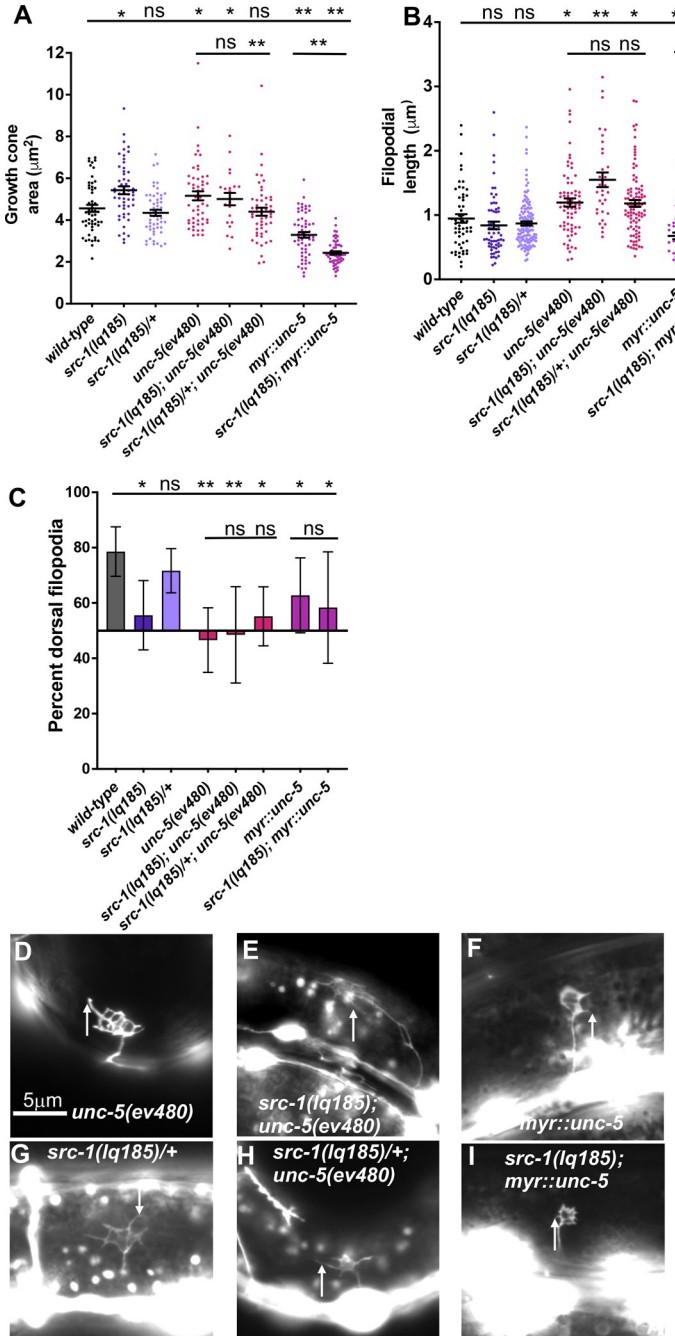

**Fig 6. VD growth cone morphological analysis in *src-1; unc-5* double mutants.** The graphs and statistical analyses in (A-C) are as described in Fig 5. At least 50 growth cones of most genotypes were analyzed, although in some genotypes fewer were analyzed because they were extremely difficult to find in the mutants (e.g. *src-1(lq185); unc-5(ev480)*). The lines indicate the genotypes analyzed for significance, with the left end of the line indicating the genotype to which significance was measured. D-I) Fluorescence micrographs of mutant VD growth cones of indicated genotypes expressing *Punc-25::gfp*. Arrows point to filopodial protrusions. Dorsal is up; anterior is left. The scale bar in (D) represents 5μm.

8A, 8B and 8F). Growth cone dorsal polarity of filopodial protrusion was also significantly reduced in *src-1(syb7248)/+* heterozygotes (Fig 8C). Heterozygous *src-1(lq185)* resembled wild-type and showed no dominant effects on axon guidance or growth cone morphology

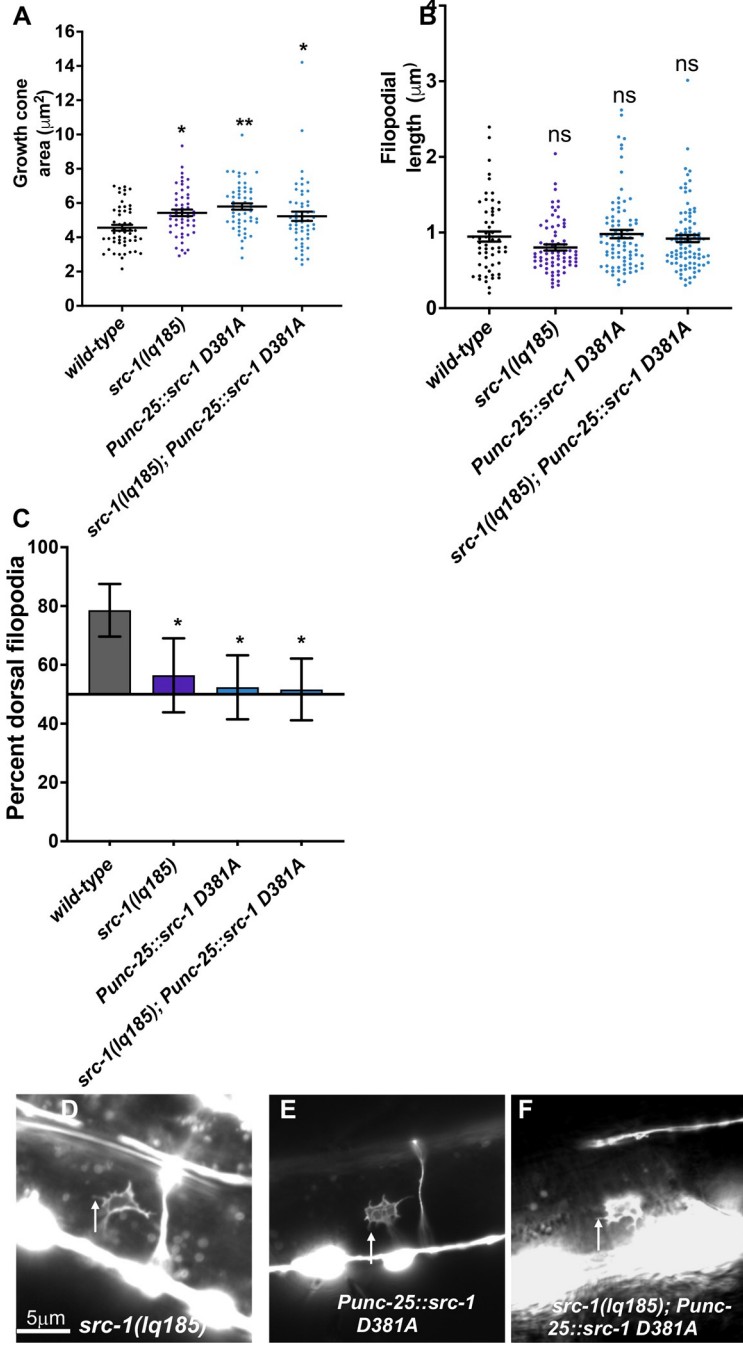

**Fig 7. VD growth cone morphological analysis of transgenic expression of kinase-dead *src-1(D381A)*.** The graphs and statistical analyses in (A-C) are as described in Fig 4. At least 50 growth cones of each genotype were analyzed. *Punc-25::src-1(D381A)* represents a transgenic line expressing the kinase-dead *src-1(D381A)*. (D-F) Fluorescence micrographs of mutant VD growth cones of indicated genotypes expressing *Punc-25::gfp*. Arrows point to filopodial protrusions. Dorsal is up; anterior is left. The scale bar in (D) represents 5μm.

(Figs 2, 3E, 8A-8C and 8E). These results indicate that the kinase activity of SRC-1 is important in axon guidance and growth cone morphology, and that kinase-dead SRC-1 molecules have a dominant-negative effect on SRC-1 function.

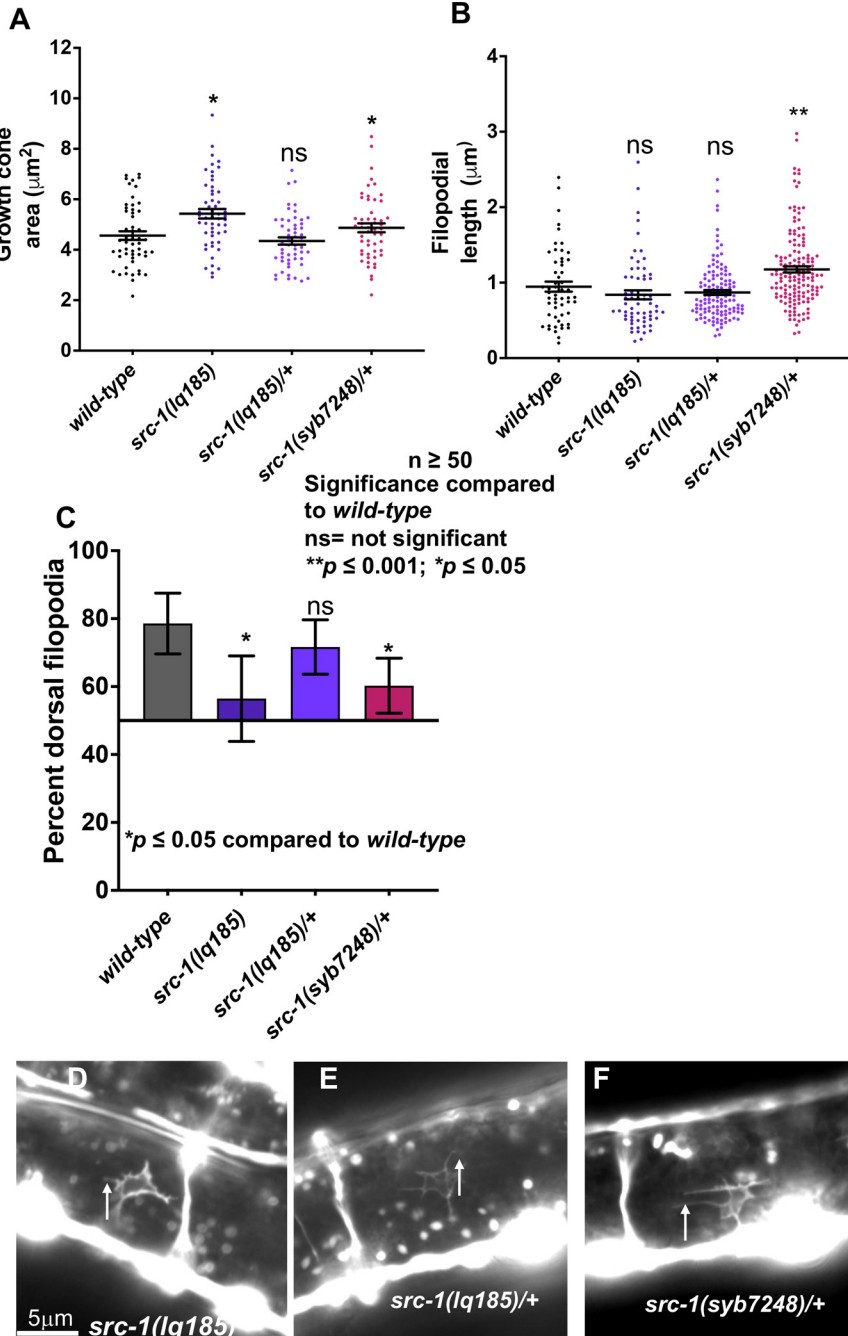

**Fig 8. VD growth cone morphological analysis of kinase-dead mutant *src-1(syb7248)*.** The graphs and statistical analyses in (A-C) are as described in Fig 4. At least 50 growth cones of each genotype were analyzed. (D-F) Fluorescence micrographs of mutant VD growth cones of indicated genotypes expressing *Punc-25::gfp*. Arrows point to filopodial protrusions. Dorsal is up; anterior is left. The scale bar in (D) represents 5μm.

## Discussion

Work presented here shows that the SRC-1 tyrosine kinase controls polarity of growth cone filopodial protrusion and extent of protrusion. *src-1* loss-of-function mutants displayed VD growth cones that were unpolarized and displayed increased growth cone area, a phenotype

similar to *unc-5* loss-of-function, consistent with previous results showing that SRC-1 and UNC-5 work together in axon guidance and cell migration [12, 13]. SRC-1 genetically acts downstream of UNC-5 in distal tip cell migration [12], and physically interacts with the cytoplasmic domain of UNC-5 [13]. Furthermore, UNC-5 cytoplasmic domain is phosphorylated on tyrosines in an UNC-6/Netrin and SRC-1-dependent manner [13, 14]. Furthermore, tyrosine 482 in the UNC-5 cytoplasmic domain is phosphorylated and is important for UNC-5 function in axon guidance and growth cone polarity and protrusion [15]. Together, these data indicate that upon UNC-6/Netrin binding, SRC-1 phosphorylates UNC-5, likely on Y482, that results in polarized VD growth cone filopodial protrusion dorsally, and inhibition of VD growth cone protrusion ventrally, leading to dorsal growth away from UNC-6/Netrin.

## A loss-of-function allele of *src-1*

When the *src-1(cj293)* mutant was sequenced using whole genome sequencing, it was found that the gene retained the potential to produce a SRC-1 molecule lacking the SH2 domain and part of the C domain regulatory region of the kinase domain, but with the SH3 domain and the catalytic portion of the kinase domain intact. Thus, *src-1(cj293)* might retain *src-1* function. Genome editing was used to precisely delete the entire *src-1* gene. *src-1(lq185)* homozygous mutants were sterile and produced no embryos. They displayed VD growth cones resembling *unc-5* mutants (unpolarized with excess growth cone area). In contrast, *src-1(cj293)* mutants were maternal-effect lethal (homozygotes produced dead embryos), suggesting that *src-1 (cj293)* retains some function relative to *src-1(lq185)*. Furthermore, *src-1(cj293)* growth cones were smaller and less-protrusive that wild-type, the opposite of *unc-5* and *src-1(lq185)* mutants. These data suggest that *src-1(cj293)* is not a complete loss-of-function and that it might retain *src-1* activity. This could be due to the production of a gain-of-function SRC-1 molecule. No cDNA representing the exon 2 to exon 5 splice was detected in *src-1(cj293)*; suggesting that the *src-1D* isoform, unaffected by the *cj293* deletion, is causing the gain-of-function phenotype.

   *src-1(lq185)* null animals grew to sterile adults, suggesting that residual maternally-provided activity is present. Thus, the complete absence of *src-1* activity might have a much stronger phenotype if both maternal and zygotic activity could be eliminated.

## SRC-1 is required cell-autonomously for VD growth cone dorsal polarity of protrusion, and to inhibit growth cone protrusion

*src-1* loss of function resulted in large, unpolarized VD growth cones similar to *unc-5* mutants. Transgenic expression of *src-1(+)* in the VD/DD neurons led to smaller, less protrusive growth cones, consistent with a normal role in inhibiting protrusion. Transgenic expression did not significantly affect growth cone polarity, but did rescue polarity defects in *src-1(lq185)* mutants. VD/DD axon guidance was also rescued. These results indicates that *src-1* is required autonomously in the VD neurons for growth cone polarity and protrusion, and the VD/DD neurons for axon guidance.

## *src-1(cj293)* might be an activated allele

The VD growth cone protrusion phenotype is opposite of that of the loss-of-function *src-1 (lq185)* phenotype. *cj293* mutants have small growth cones, and *lq185* mutants have large growth cones. This suggests that *cj293* might have excess *src-1* activity and might be an activated allele. The SH2 domain is missing in *cj293*, as is regulatory region of the kinase domain containing K290 shown to be important for SRC-1 function [13]. Possibly, lack of one or both of these features leads to excess SRC-1 activity in inhibiting protrusion, although the SH2

domain is required for interaction with the UNC-5 cytoplasmic domain [13]. *src-1(+)* trans-genic expression also inhibited growth cone area, consistent with overactivity. However, trans-genic *src-1(+)* expression did not affect growth cone polarity as did *cj293*. Possibly, *cj293* has distinct effects on polarity and protrusion. A full analysis of the characteristics of *cj293* is beyond the scope of this report.

### Aspartate 381 is important for SRC-1 function

Aspartate 381, present in the catalytic pocket of the kinase domain, was altered to alanine in *src-1(+)* transgenes (D381A). *src-1(D381A)* expression alone caused severe VD/DD axon guid-ance defects, increased VD growth cone area, and loss of VD growth cone polarity, similar to *src-1(lq185)* loss-of-function. *src-1(D381A)* failed to rescue any aspect of the *src-1(lq185)* phe-notype. This indicates that D381 is important for SRC-1 function, and that expression in a wild-type background results in a dominant-negative phenotype.

A D381A mutation was produced in the endogenous *src-1* locus by genome editing. *src-1 (syb7248)* resulted in larval lethality, unlike the sterile adults produced by *src-1(lq185)*. This indicates that *src-1(syb7248)* had a stronger effect on viability than *src-1(lq185)*. VD growth cones could not be identified in *src-1(syb7248)* arrested larvae. However, heterozygous *src-1 (syb7248)* animals displayed increased VD growth cone area and loss of polarity, suggesting that it has a dominant negative effect. Possibly, SRC-1(D381A) interacts with targets but fails to phosphorylate them, thus removing the target molecules, such as UNC-5, from acting in the process that they would normally control.

### SRC-1 and UNC-5 in axon growth cone polarity and protrusion

Previous studies implicate SRC-1 acting with UNC-5 in axon guidance and cell migration, and UNC-5 being a target of SRC-1 tyrosine phosphorylation [12–14]. UNC-5 Y482 is important in axon guidance and VD growth polarity and protrusion and is phosphorylated [15] (Maha-dik and Lundquist, 2023), likely by SRC-1. VD growth cones of double mutants of *unc-5* and *src-1* loss-of-function resembled *unc-5* and *src-1* mutants alone, consistent with them acting in the same pathway. However, *src-1* was not required for the inhibitory effects of constitutively-active *myr::unc-5*. This suggests that SRC-1 does not act downstream of UNC-5. It is also possi-ble that SRC-1 plays a role in UNC-5 receptor dimerization upon activation by UNC-6/Netrin (Fig 9), which might be bypassed by MYR::UNC-5, which does not require UNC-6/Netrin binding for activation.

## Materials and methods

### Genetic methods

Experiments were performed at 20˚C using standard *C. elegans* techniques [23]. Mutations used were as follows. LGI: *src-1(lq185)*, *src-1(cj293)*, *src-1(syb7248)*, *tmC20*; LGII: *juIs76* [*Punc-25::gfp*]; LGIV: *unc-5(e791, ev480)*. The presence of mutations was confirmed by phenotype and sequencing. Chromosomal locations not determined: *lqIs296[Punc-25::myr::unc-5::gfp]; lqIs377 [Punc-25::src-1::gfp], lqIs394 [Punc-25::src-1(D381A)]*.

As *src-1* mutations were sterile or maternal effect lethal, all *src-1* homozygotes scored had wild-type maternal *src-1* activity. *src-1* mutations were balanced using the balancer chromo-some *tmC20* [24].

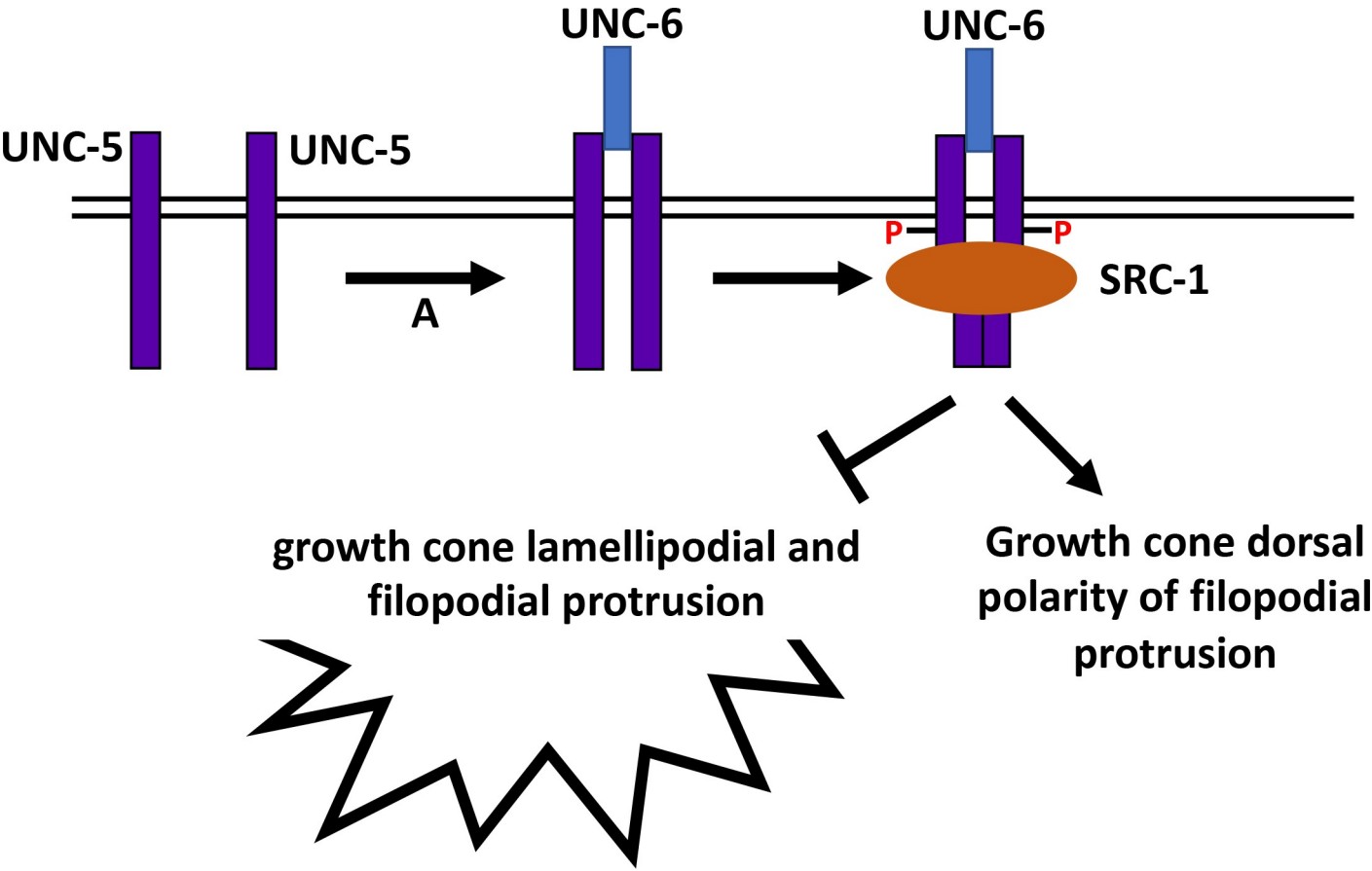

**Fig 9. A model of SRC-1 and UNC-5 in VD growth cone polarity and protrusion.** This model incorporates work presented here as well as in previous studies. UNC-6 interaction with UNC-5 results in receptor dimerization (A). UNC-5 dimers then serve as a substrate for SRC-1 phosphorylation of Y482 and UNC-5 cytoplasmic domain association, which results in growth cone dorsal polarity of protrusion and inhibition of protrusion ventrally, resulting in net dorsal outgrowth.

### Sequencing of the *src-1(cj293)* deletion

*src-1(cj293)* genomic DNA was sequenced using single read, 150-cycle Illumina sequencing. Reads were aligned to the reference genome using BWA-MEM2 [25, 26]. Resulting BAM files were analyzed in the Integrated Genome Viewer [27, 28] (Fig 1C). Two representative reads spanning the *cj293* deletion region are shown in S1 File. Sequence data can be accessed on the Sequence Read Archive, BioProject ID PRJNA973980.

### RT-PCR

Total RNA from N2 and *src-1(cj293)* was isolated as described previously [29]. RT-PCR was conducted using the Superscript^tm IV One-Step RT-PCR system (Invitrogen, Waltham, MA, USA). The following sequences were used to amplify the wild-type *scr-1A cDNA*, with PstI and XbaI sites on the ends, respectively:

src-1F (forward): `ctgcagATGGGTTGCCTGTTTTCAAAA`
src-1RT (RT and reverse): `tctagaGGCACTTGGTGGCGCGTAATT`

The following sequences were used to detect the potential *src-1(cj293)* exon 2 to exon 5 splice:

cj293RTF (forward): `AACCATCTCTCAACTAAACGG`

cj293RT (RT and reverse): `TCGTAAATGTCCTCTTCCATC`

## Transgene construction

*Punc-25*::*src-1*:P:*gfp* was made by amplifying the *src-1A* cDNA by RT-PCR and placing this behind the *unc-25* promoter (the pEL1135 plasmid). *Punc-25*::*src-1(D381A)*::*gfp* was created by site-directed mutagenesis by inverse PCR using the *Punc-25*::*src-1*::*gfp* construct (the pEL1169 plasmid). The coding regions for both transgenes were sequenced to ensure no errors had been introduced by PCR. The sequences of these plasmids can be found in Supplemental Material.

## Cas9 genome editing to generate *src-1(lq185)* and *src-1(syb7248)D381A*

CRISPR-Cas9 genome editing was used to precisely delete the entire gene of *src-1* (Fig 1) (see S1 File for the sequences of guide RNAs, the single-stranded oligonucleotide repair template, and sequence of the *lq185* mutation). Synthetic guide RNAs were directed at the 5' and 3' ends of the coding region of the gene. A mix of sgRNAs, a single stranded oligonucleotide repair template, and Cas9 enzyme was injected into the gonads of N2 animals, along with the *dpy-10 (cn64)* co-CRISPR reagents [30]. Deletion of *src-1* was confirmed by PCR and sequencing. Genome editing reagents were produced by InVivo Biosystems (Eugene, OR, USA).

CRISPR-Cas9 genome editing was used to alter the codon for aspartate 381, changing it to alanine. The sequence of *src-1(syb7248)* is in S1 File, and was confirmed by PCR and sequencing. *src-1(syb7248)* was produced by SUNY Biotech (Fuzhou, China).

## Quantification of axon guidance defects

VD/DD neurons were visualized with the *Punc-25*::*gfp* transgene *juIs76* [31], which is expressed in GABAergic motor neurons including 13VDs and 6DDs. Axon guidance defects were scored as previously described [21]. In wild type, an average of 16 of the 19 commissures of VD/DD axons are distinguishable, as axons can be present in a fascicle and thus cannot be resolved. A total of 100 animals were scored (1600 total commissural processes). Total axon guidance defects were calculated by counting all the axons which failed to reach the dorsal nerve cord, wandered at an angle of 45˚ or greater, crossed over other processes, or displayed ectopic branching. Significance of difference between two genotypes was determined using Fisher's exact test.

## Growth cone morphological imaging and quantification

VD growth cones were imaged as previously described [4, 9, 18–21]. L1 to late L2 larval animals were harvested 16-hour post-hatching at 20˚C and placed on a 2% agarose pad with 5mM sodium azide in M9 buffer. Compared to the wild-type, mutant alleles showed the slower emergence of VD/DD neurons for ventral nerve cord, in turn delaying the extension of VD growth cones. Hence, the time point of 16-hour post -hatching is modified depending on the extension time of VD neurons. For example, 21 hours was the time of VD growth cone emergence for *unc-5* mutants.

Growth cones were imaged with a Qimaging Retiga EXi camera on a Leica DM5500 microscope at 100x magnification. Projections less than 0.5um in width were scored as filopodia. Growth cone area and filopodial length were quantified using ImageJ software. Quantification was done as described previously [4, 9, 18–21]. Significance of difference between two genotypes was determined by two-sided *t*-test with unequal variance.

Polarity of growth filopodial protrusions was determined as previously described [4, 9, 18–21]. Growth cone images were divided into two halves, dorsal and ventral, with respect to the ventral nerve cord. The number of filopodia in each half was counted. The proportion of dorsal filopodia was determined by the number of dorsal filopodia divided by total number of filopodia. Significance of difference between two genotypes was determined by Fisher's exact test. In most cases, at least 50 growth cones were scored, although in some genotypes fewer were scored due to the difficulty in finding any VD growth cones that emerged from the ventral cord.

## Supporting information

**S1 File. Sequences of Cas9 genome editing used to create *src-1(lq185)* and *src-1(syb7248)*.**
(DOCX)

**S2 File. The sequences of the *Punc-25::src-1A::gfp* and *Punc-25::src-1A(D381A)* plasmids pEL1135 and pEL1169.**
(DOCX)

**S3 File. The sequences of the *Punc-25::src-1A::gfp* and *Punc-25::src-1A(D381A)* plasmids pEL1135 and pEL1169.**
(DOCX)

## Acknowledgments

The authors thank E. Struckhoff, Z. Grant, and C. McKimens for technical assistance. Some strains were provided by the CGC, which is funded by NIH Office of Research Infrastructure Programs. Sequencing was conducted at the KU Genome Sequencing Core.

## Author Contributions

**Conceptualization:** Erik A. Lundquist.

**Data curation:** Snehal S. Mahadik, Erik A. Lundquist.

**Formal analysis:** Snehal S. Mahadik, Emily K. Burt.

**Investigation:** Snehal S. Mahadik, Emily K. Burt.

**Methodology:** Snehal S. Mahadik.

**Project administration:** Erik A. Lundquist.

**Supervision:** Erik A. Lundquist.

**Validation:** Snehal S. Mahadik.

**Visualization:** Snehal S. Mahadik.

**Writing – original draft:** Snehal S. Mahadik.

**Writing – review & editing:** Erik A. Lundquist.

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
