## [Decision Letter · Decision Letter 0]

30 Oct 2023

PONE-D-23-28317SRC-1 controls growth cone polarity and protrusion with the UNC-6/Netrin receptor UNC-5 in Caenorhabditis elegansPLOS ONE

Dear Dr. Lundquist,

Thank you for submitting your manuscript to PLOS ONE. As you can see, the comments of the reviewers are generally quite positive. However, Reviewer 2 makes the technical suggestion that the authors should verify by RT-PCR the existence of the novel SRC splice variant whose presence they postulate to account for the phenotype of the mutant allele cj293. This seems to me to be a sensible suggestion. Demonstrating the presence of this variant RNA would materially strengthen the authors’ argument; should they be unable to detect it, it would be appropriate for them to comment on possible reasons for that. This is considered a **minor revision that will not require re-review** regardless of the outcome of the experiment. If there is some reason why this experiment is not practical, it would also be acceptable for the authors to submit a revision simply pointing out the issue that the reviewer raises and explaining why it cannot be tested readily. Therefore, we invite you to submit a revised version of the manuscript that addresses this point raised during the review process. I should also say that I do apologize for the time that was required for this review process. In addition to the now usual challenges of arranging referees there were some unfortunate miscommunications between myself, the editorial office, and the reviewers, for which I must take the responsibility, and again, offer my regrets. 

We look forward to receiving your revised manuscript.

Kind regards,

Edward Giniger

Academic Editor

PLOS ONE

Journal Requirements:

EAL, R03NS114554, National Institute of neurological Disorders and Stroke

EAL, R01NS115467, National Institute of neurological Disorders and Stroke

PI not an author, P30GM145499, National Institute of General Medical Sciences

PI not an author, P20GM103418, National Institute of General Medical Sciences

PI not an author, P40OD010440, National Institutes of Health

The authors thank E. Struckhoff, Z. Grant, and C. McKimens for technical assistance. Some strains were provided by the CGC, which is funded by NIH Office of Research Infrastructure Programs (P40OD010440). Sequencing was conducted at the KU Genome Sequencing Core supported by the National Institute of General Medical Sciences Center for Molecular Analysis of Disease Pathways (P30GM145499). The Kansas Infrastructure Network of Biomedical Research Excellence (P20GM103418) provided computational support. This work was supported by National Institutes of Neurological Disorders and Stroke grants R03NS114554 and R01NS115467 to E.A.L.

EAL, R03NS114554, National Institute of neurological Disorders and Stroke

EAL, R01NS115467, National Institute of neurological Disorders and Stroke

PI not an author, P30GM145499, National Institute of General Medical Sciences

PI not an author, P20GM103418, National Institute of General Medical Sciences

PI not an author, P40OD010440, National Institutes of Health

7. Please upload a copy of Figure 9, to which you refer in your text. If the figure is no longer to be included as part of the submission please remove all reference to it within the text.

Reviewers' comments:

Reviewer's Responses to Questions

**Comments to the Author**

1. Is the manuscript technically sound, and do the data support the conclusions?

Reviewer #1: Yes

Reviewer #2: Partly

2. Has the statistical analysis been performed appropriately and rigorously? 

Reviewer #1: Yes

Reviewer #2: I Don't Know

3. Have the authors made all data underlying the findings in their manuscript fully available?

Reviewer #1: Yes

Reviewer #2: Yes

4. Is the manuscript presented in an intelligible fashion and written in standard English?

Reviewer #1: Yes

Reviewer #2: Yes

5. Review Comments to the Author

Reviewer #1: I was not an original reviewer of this article. However, from my reading and assessment, Authors have addressed reviewer concerns, except confirming the expression of the older src mutant. Based on the genomic sequencing and difference in phenotyping, I am ok with this.

Reviewer #2: The Src family kinase SRC-1 has been previously demonstrated to interact physically with UNC-5 and to phosphorylate UNC-5, to guide axons. Mahadik et al. now show that SRC-1 controls VD growth cone polarity and protrusion with UNC-5 in nematodes. They generated a precise deletion of src-1 and showed that these mutants have unpolarized growth cones displaying an increase in size, a phenotype similar to unc-5 mutants. Through src-1 rescue, they show that this effect is cell autonomous. They also generated a kinase dead src-1 allele (D831A), which likely has dominant negative activity. Their genetic interaction experiments using src- 1 and unc-5 suggest that they mostly act in the same pathway on growth cone polarity and protrusion. The phenotype of an activated unc-5 (myr::unc-5) in absence of src-1 suggests that src-1 might act upstream of UNC-5 and modulate UNC-5 dimerization.

This work is novel in the analysis of SRC-1 in growth cone morphology and the connection with UNC-5 in this process. The work is well done and well quantified. However, I have one major comment that should be addressed:

The authors sequenced a previously generated SRC-1 allele harboring a deletion and they state, “if the fusion intron consisting of the 5’ splice site of exon two and 3’ splice site of exon four was removed from transcripts, and exons two and five were spliced together, the src-1 open reading frame would be retained.” From this, they further predict that a mutant SRC-1 protein would be produced. As this is purely speculative but important for their interpretation of the function of this allele (cj293) in growth cones, they need to show that it exists at the mRNA level by RT-PCR and sequencing. This is key for their interpretation as absence of mRNA production would significantly affect the interpretation of the src-1(cj293) allele. Of note, I am not asking here for biochemical experiments, I am simply asking to check that the mRNA that the authors predict is indeed generated. This is a straightforward experiment that shouldn’t take very long to perform.

6. PLOS authors have the option to publish the peer review history of their article (what does this mean?). If published, this will include your full peer review and any attached files.

Reviewer #1: No

Reviewer #2: No

---

## [Author Response · Author response to Decision Letter 0]

22 Nov 2023

Review Comments to the Author

Reviewer #1: I was not an original reviewer of this article. However, from my reading and assessment, Authors have addressed reviewer concerns, except confirming the expression of the older src mutant. Based on the genomic sequencing and difference in phenotyping, I am ok with this.

Reviewer #2: The Src family kinase SRC-1 has been previously demonstrated to interact physically with UNC-5 and to phosphorylate UNC-5, to guide axons. Mahadik et al. now show that SRC-1 controls VD growth cone polarity and protrusion with UNC-5 in nematodes. They generated a precise deletion of src-1 and showed that these mutants have unpolarized growth cones displaying an increase in size, a phenotype similar to unc-5 mutants. Through src-1 rescue, they show that this effect is cell autonomous. They also generated a kinase dead src-1 allele (D831A), which likely has dominant negative activity. Their genetic interaction experiments using src- 1 and unc-5 suggest that they mostly act in the same pathway on growth cone polarity and protrusion. The phenotype of an activated unc-5 (myr::unc-5) in absence of src-1 suggests that src-1 might act upstream of UNC-5 and modulate UNC-5 dimerization.

This work is novel in the analysis of SRC-1 in growth cone morphology and the connection with UNC-5 in this process. The work is well done and well quantified. However, I have one major comment that should be addressed:

The authors sequenced a previously generated SRC-1 allele harboring a deletion and they state, “if the fusion intron consisting of the 5’ splice site of exon two and 3’ splice site of exon four was removed from transcripts, and exons two and five were spliced together, the src-1 open reading frame would be retained.” From this, they further predict that a mutant SRC-1 protein would be produced. As this is purely speculative but important for their interpretation of the function of this allele (cj293) in growth cones, they need to show that it exists at the mRNA level by RT-PCR and sequencing. This is key for their interpretation as absence of mRNA production would significantly affect the interpretation of the src-1(cj293) allele. Of note, I am not asking here for biochemical experiments, I am simply asking to check that the mRNA that the authors predict is indeed generated. This is a straightforward experiment that shouldn’t take very long to perform.

This is a good point the reviewer makes. We tried several times to amplify a cDNA from src-1(cj293) spanning the predicted exon 2 to exon 5 splice and were unsuccessful. Possibly this splice does not occur. On Wormbase, a src-1D isoform is described consisting of only the last two exons encoding the kinase domain. cj293 does not affect the coding region of predicted src-1D isoform. Possibly, production of src-1D without the other isoforms in cj293 results in the gain-of-function phenotype. We now mention this as an alternate possibility in the manuscript.

---

## [Editor Report · Decision Letter 1]

28 Nov 2023

SRC-1 controls growth cone polarity and protrusion with the UNC-6/Netrin receptor UNC-5 in Caenorhabditis elegans

PONE-D-23-28317R1

Dear Dr. Lundquist,

We’re pleased to inform you that your manuscript has been judged scientifically suitable for publication and will be formally accepted for publication once it meets all outstanding technical requirements.

Kind regards,

Edward Giniger

Academic Editor

PLOS ONE
---

## [Editor Report · Acceptance letter]

6 Dec 2023

PONE-D-23-28317R1 

SRC-1 controls growth cone polarity and protrusion with the UNC-6/Netrin receptor UNC-5 in *Caenorhabditis elegans*

Dear Dr. Lundquist:

I'm pleased to inform you that your manuscript has been deemed suitable for publication in PLOS ONE. Congratulations! Your manuscript is now with our production department. 

Kind regards, 

on behalf of

Dr. Edward Giniger 

Academic Editor

PLOS ONE